# Sulfonation Reactions behind the Fate of White Wine’s Shelf-Life

**DOI:** 10.3390/metabo12040323

**Published:** 2022-04-02

**Authors:** Maria Nikolantonaki, Rémy Romanet, Marianna Lucio, Philippe Schmitt-Kopplin, Régis Gougeon

**Affiliations:** 1Institut Universitaire de la Vigne et du Vin, UMR PAM Université de Bourgogne/Institut Agro Dijon, Jules Guyot, Rue Claude Ladrey, BP 27877, 21078 Dijon, France; remy.romanet@u-bourgogne.fr (R.R.); regis.gougeon@u-bourgogne.fr (R.G.); 2DIVVA (Développement Innovation Vigne Vin Aliments) Platform/PAM UMR, IUVV, Rue Claude Ladrey, BP 27877, CEDEX, 21078 Dijon, France; 3Research Unit Analytical BioGeoChemistry, Helmholtz Zentrum Muenchen, 85764 Neuherberg, Germany; marianna.lucio@helmholtz-muenchen.de (M.L.); schmitt-kopplin@helmholtz-muenchen.de (P.S.-K.); 4Analytical Food Chemistry, Technische Universität München, Alte Akademie 10, 85354 Freising, Germany

**Keywords:** chardonnay, oxidation, oxidative stability, peptides, phenolic compounds, glutathione

## Abstract

White wine’s oxidative stability after several years of bottle aging is synonymous to its organoleptic quality. In order to gain control over the cascade of chemical reactions that are implicated in that phenomenon, fourier transform ion cyclotron resonance mass spectrometry (FT-ICR-MS)-based metabolomics and sensory evaluation were combined for the analysis of a vertical series of white wines from different vineyard plots. Data mining using supervised cluster analysis allowed the extraction of known and unknown sulfur- and nitrogen-containing molecular features, with oxidative stability molecular markers presenting an increased number of S and O atoms in their formulas. In their majority, S-containing molecular features possessed between 4 to ~12 O atoms, indicating the relatively higher importance of sulfonation reactions as opposed to dimerization reactions. Molecular networking, based on sulfonation reaction transformations, evidences the importance of hitherto unknown and/or minor sulfur dioxide binders (peptides, aldehydes, and polyphenols) on wine’s oxidative stability.

## 1. Introduction

During bottle aging, wine is exposed to relatively low quantities of oxygen that are nevertheless sufficient to influence the outcomes of their shelf-life period. In particular, reactive oxygen species (ROS), formed by the reduction of molecular oxygen, modulate the extent of different reactions involving volatile and nonvolatile components, resulting in the formation and/or degradation of a number of powerful aroma compounds, with major consequences on the process of aromas’ oxidative evolution [1]. At this time, numerous authors have already reported studies of the mechanisms involved in wine oxidation, and many of them were targeted at phenolic compounds, considered to be primary substrates for oxidation [2]. However, other chemical reactions taking place during bottle aging do not involve oxygen, meaning that, even in an environment completely devoid of oxygen, a certain form of aging will occur [3].

Above all, these results emphasized the dependency of the oxygenation process on the wine’s compositions, where different wines could exhibit different reactivity towards oxygen, thus stressing the need for analytical approaches that can provide comprehensive pictures of subtle mechanistic variations among series of samples. In that respect, non-targeted mass spectrometry-based metabolomics have recently shown great potential in describing the evolution of chemical spaces involved in enological practices [4,5,6,7], but very few non-targeted studies have actually analyzed the evolution of wine during bottle aging [8,9,10,11] and even fewer have done so in relation to its oxidative evolution [12]. A recent study from our research group showed that controlled electrochemical oxidation of chardonnay wines could reveal the diversity of sulfur-containing compounds, which are most sensitive to oxidative degradation, and the nitrogen-containing compounds resulting from oxidation [13]. Moreover, the application of a multidisciplinary approach that combined sensory evaluation, chemical and metabolomic analyses, and investigating the oxygen transfer through the bottleneck/stopper elucidated the importance of the glass/cork interface [12]. It shows unambiguously that the transfer of oxygen at the interface between the cork stopper and the glass bottleneck must be considered a potentially significant contributor to the oxidation state during bottle aging, leading to a notable modification of a wine’s sulfur–nitrogen chemical signature.

Sulfur- and nitrogen-containing compounds constitute the antioxidant molecular capital of white wines in the bottle and determine their aging potential [14,15]. These compounds are grape, yeast, and bacteria metabolism derivatives and act as natural antioxidants while reacting as sacrificial nucleophiles with quinones and protecting wines against oxidative spoilage [16,17,18,19]. In addition, sulfites are present in the most widely used additive as antioxidant in winemaking. High-resolution mass spectrometry-based metabolomic studies could easily fingerprint wines according to the SO_2_ concentration added to must at pressing or to wine at bottling [20,21]. These results thus bring unprecedented molecular scale information about the sulfur-related chemistry that is involved immediately at the beginning of the winemaking process in the must and that is carried out throughout bottle aging. It is shown that a great diversity of metabolite classes are actually impacted by sulfur dioxide additions to must or wine, including amino acids, carbohydrates, phenolic compounds, and indoles. S-sulfonated products of flavanols, indoles, and amino acids have been recently reported in bottle-aged wines from different cultivars, while the physicochemical parameters controlling sulfonation reactions remain today poorly understood [21,22].

All these studies highlighted the extent to which the chemistry that could be involved in wine’s oxidative stability during bottle aging is still unknown. The main scope of the present study was to examine the importance of key chemical reactions such as sulfonation in wine’s oxidative stability during bottle aging. In particular, in this work, we aimed to compare the metabolic profile of white wines, sealed with a natural cork, sensory evaluated as oxidized or good (not oxidized) after several years of bottle aging under cellar conditions. To obtain a high level of variability, the experimental design included wines from the same winery from two vintages and five vineyard plots. An investigation of known and unknown sulfonation reaction products was performed by means of in silico deconjugation.

## 2. Results and Discussion

### 2.1. Multivariate FT-ICR-MS-Based Statistical Analysis of Wines

Unsupervised PCA was initially conducted to obtain an overview of FT-ICR-MS data from wines exhibiting oxidative (O) or not (G) sensory characters. As shown in Figure 1, samples were mostly discriminated according to the vintage along with the first component and according to plots along with the second component. Interestingly, for each plot within a given vintage, G and O samples appeared grouped, although this was to a lesser extent for the 2005 vintage. Yet, to maximize the discrimination among the classes and explore the potential molecular marker candidates in detail, hierarchical cluster analysis readily revealed G and O subclusters within each vintage (Figure 2A). The analysis readily revealed two distinct clusters of O and G wines, each including the two vintages. The extensive chemical dissimilarity between the two main clusters (O and G) is depicted in Figure 2B,C. The van Krevelen diagrams and ring charts display the most representative molecular formulas (correlation coefficient *r* > 0.8) for each cluster. G-related molecular markers were dominated by sulfur-containing molecules (CHOS and CHONS) with contributions to the total intensity of assigned molecular formulas of up to 55% and 39%, respectively. On the other hand, O-related molecular markers were characterized by a significantly higher contribution of CHON formulas with 31% of the total intensity of assigned molecular formulas (28% for G) and lower contributions of sulfur-containing formulas, in particular CHOS (27% and 28% for CHOS and CHONS respectively). This result is in accordance with recent published data reporting the consumption of CHOS and the formation of CHON and CHONS compounds after controlled electrochemical simulation of oxidative reactions in white wine [13]. Annotated elemental formulas from identified discriminant mass peaks revealed that markers for G wines are found in the areas of sulfonated polyphenols and amino acids/peptides [23]. However, O-related markers appear mostly located in the area of polyphenols (Figure 2B,C).

The van Krevelen diagrams (H/C versus O/C) were generally used to identify the structural properties of molecular markers in FT-ICR-MS research, as only the molecular formula was given. Upon comparing the ratios of H/C and O/C for CHOS- and CHONS-containing compounds, products on the right-hand side of the van Krevelen diagram with higher O/C ratios, G samples allowed the highest number of oxygen atoms to support sulfonation (-SO_3_H) and nitration (-NO_3_H) reactions, resulting good markers (Figure 2). In addition to van Krevelen diagrams, plotting elemental contributions as bar charts also provided useful overview analyses. In particular, comparing the oxygen class distributions of G and O groups of molecular markers gave useful additional information. Figure 3A shows that the number of CHO formulas bearing 6 to 11 O atoms increased sharply with oxidation. However, the number of molecular formulae containing up to 2 atoms of sulfur (S) was higher in G samples (Figure 3B). Moreover, in their majority, S-containing molecular features possessing between 4 to ~12 O atoms appeared to be favored for the G state of aged wines, indicating the relatively higher importance of sulfonation reactions as opposed to dimerization reactions, which is linked to wine’s oxidative stability during aging (Figure 3C).

### 2.2. Compositional Networks for Metabolite Identification

In order to give more insights into the chemical mechanisms implicated in wine’s oxidative stability during bottle aging, an alternative approach exploiting the exact mass information provided by FT-ICR-MS through the building of compositional networks was applied in the present study. Compositional networks enable mass coverage beyond classical annotations and thus the identification of unknowns. Exact mass from FT-ICR-MS data thus provides the chemical basis on which we can construct sample-related metabolic pathways in a data-driven network-based approach. By displaying data points as nodes that conditionally become connected (edges) to each other, networks have been proven ideal for efficient modeling and visualization of multivariate data [24]. The total network of G markers appeared clustered according to chemical families, where CHONS-containing compounds (in red) and CHOS (in green) were present in condensed areas of the network (Figure 4). However, CHO- (in blue) and CHON- (in orange) containing compounds were distributed across the network. Using mass difference network analysis, it became possible to make assumptions about the structure of yet-unknown markers, depending on the identity of their *m/z* connectivity. For the present study, connections (edges) were referred to mass differences specific to nucleophilic addition reactions of some chemical families, including known phenols, sulfites (HSO_3_^−^), carbonyls, and amino acids—peptide compounds with unknowns in G-related molecular markers. In the presented compositional network, edges are colored according to the chemical families mentioned above. The clustering of G markers shows clearly that CHONS-containing compounds resulted from the sulfonation reaction essentially of amino acids and peptides, while CHOS-containing compounds are the reaction products of essentially polyphenol and carbonyl sulfonation reactions. Among connected masses, 172 were connected by targeted grape- and oak wood-derived polyphenols, 504 were connected to carbonyls, 310 were connected to sulfites, and 145 were connected to amino acid and peptides. In all targeted groups of chemical families, some of the selected masses exhibited very high connectivity (>10), thus suggesting its key role in wine’s oxidative stability during bottle aging. In detail, among targeted polyphenols, globally non-carboxyl monophenols, tyrosol, and hydroxytyrosol (formed during yeast fermentation from tyrosine), followed by main grape-derived phenolic acids (caffeic, ferulic, and coumaric acid) showed higher connectivity than flavanols (catechin and epicatechin).

Sulfonation reactions occur under wine’s acidic conditions from the very beginning of winemaking until/during bottle aging. In wine matrices (pH = 3.0–4.0), sulfites are present in the form of bisulfite ions (HSO_3_^−^), which possibly react with wine-relevant nucleophiles, mainly polyphenols, amino acids, sugars, keto acids, and other unknown compounds. The reactivity/behavior of HSO_3_^−^ changes according to the metabolic environment, which changes constantly during aging. The few known sulfite binders are just the tip of the iceberg, and their presence can only partially explain the variability related to the binding strength of sulfites on different wines [25]. Indeed, sulfonation reactions, even for compounds known as strong binders such as acetaldehyde, can be reversible [26], indicating the potential release of free sulfur dioxide during bottle aging. This result highlights the importance of identifying the hitherto-unknown and minor binders in order to better estimate the antioxidant capacity of bound sulfur dioxide. In the present study, 310 links were found between annotated peaks from global wine FT-ICR-MS data on the basis of sulfonation reaction transformations. The mass difference corresponds exactly to the mass difference between the initial targeted compound and the sulfonation product.

Further investigations were conducted to putatively identify the compounds corresponding to sulfonation products. The mass differences (Δppm) between measured and exact theoretical masses were calculated to support the compound identification. A total of 6 compounds corresponding to 164 nodes from the network were putatively identified as known analogues of polyphenols, carbonyls, and amino acids—peptide compounds with a very good error (Δppm < 1). Indeed, the unidentified masses at *m/z* 217.0175; *m/z* 273.0074; *m/z* 369.0284; *m/z* 124.9914; *m/z* 305.0812; and *m/z* 386.0332 assigned to [C_8_H_10_O_5_S]^−^; [C_10_H_10_O_7_S]^−^; [C_15_H_14_O_9_S]^−^; [C_2_H_6_O_4_S]^−^; [C_2_H_6_O_4_S]^−^; [C_11_H_18_O_6_N_2_S]^−^; and [C_10_H_17_O_9_N_3_S_2_]^−^ formulae could be identified as tyrosol, ferulic acid, catechin, acetaldehyde, γ-glutamyl-cysteine, and glutathione sulfonation products, respectively. In general, the sulfonation products of the above-mentioned compounds were quantified in all analyzed wines from 2004 and 2005 vintages, and the box plots of their relative abundances are reported in Figure 5. All 6 sulfonation products were detected in all wines, presenting relative abundances in a vintage-dependent manner. It should be pointed out that tyrosol, acetaldehyde, and catechin sulfones were significantly more abundant in wines from the 2005 vintage, whereas sulfonated glutathione was more abundant in wines from the 2004 vintage. No significant difference was observed for ferulic acid and γ-glutamyl-cysteine derivatives. Considering the quantitative ratio among analyzed vintages between the sulfonated polyphenols (tyrosol and catechin) and the glutathione, we can formulate the hypothesis that sulfonation reactions are highly dependent on the chemical environment as well as other physicochemical parameters (i.e., oxygen intake, wine pH) that occur during bottle aging.

## 3. Materials and Methods

### 3.1. Chemicals

MS-grade methanol was obtained from Fisher Scientific (Loughborough, UK) and ultrapure water (18.2 MΩ·cm) was obtained from a Milli-Q system (Merck; Darmstadt, Germany).

### 3.2. Experimental Design

The experimental design included 20 chardonnay wines from five different vineyard plots located in Burgundy—North East France, from vintages of 2004 and 2005, produced according to standard winemaking procedures (grapes were destemmed and crushed before being put into tanks; alcoholic fermentation was brought to dryness and followed by malolactic fermentation) and aged in oak wood barrels for nine months at the same winery. Wines were bottled after finning using natural cork and aged in cellar conditions. They were all analyzed in June 2015.

### 3.3. Sensory Analysis

A panel of six trained judges performed sensory analyses of the wines. All panelists had extensive experience in wine tasting and participated regularly in sensory panels with chardonnay wines. All of the assessments were performed at room temperature (18 ± 1 °C) in individual booths under daylight lighting. Fifty milliliters of wine were poured in standard ISO 3591 XL5-type tasting glasses with glass covers identified by three-digit random codes and assessed within 15 min of pouring. Each wine was submitted to the panelists just after the bottle was opened. They were asked to evaluate the presence or absence of oxidation flavors (G: no oxidative character; O: oxidative character recognized).

### 3.4. Fourier Transform Ion Cyclotron Resonance Mass Spectrometry Analysis and Data Processing

Analyses were realized using a Bruker SolariX Ion Cyclotron Resonance Fourier Transform Mass Spectrometer (FT-ICR-MS) (BrukerDaltonics GmbH; Bremen, Germany) equipped with a 12 Tesla superconducting magnet (Magnex Scientific Inc.; Yarnton, GB, USA) and an APOLO II ESI source (BrukerDaltonics GmbH; Bremen, Germany) in negative ion mode. Samples were diluted at 1/20 into methanol and injected at flow rate of 120 µL/h into the electrospray. The broadband spectra were acquired between 147 *m/z* and 1000 *m/z* mass range in 2 MW FIDs and summed over 300 scans. All samples were analyzed randomly. The MS was first calibrated using arginine ion clusters (57 nmol/mL in methanol). Next, raw spectra were further internally calibrated using a reference list including known wine markers and ubiquitous fatty acids to achieve the best possible mass accuracy and precision among the samples [4].

### 3.5. Data Mining

Evaluation of statistical significance was conducted by one-way analysis of variance (ANOVA) followed, if significant (*p* < 0.05), by a posteriori Duncan test. Differences between groups were considered significant when *p* < 0.05. Principal Component Analysis (PCA) and Hierarchical Cluster Analysis (HCA) were obtained with Perseus software (v.1.5.1.6 http://www.perseus-framework.org, accessed on 1 January 2022, Max Planck Institute of Biochemistry, Germany) [27]. The clustering was performed using a Pearson’s correlation. FT-ICR-MS data were analyzed with DataAnalysis (v.4.3, Bruker Daltonik GmbH; Mannheim, Germany). Features (*m/z* peaks) were filtered according to S/N higher than 10 and an absolute intensity above 2 × 10^6^. Features alignment was made with the Matrix Generator software (v.0.4, Helmholtz Zentrum Muenchen) with a mass accuracy window of 1 ppm and filtered to keep features present in 90% of samples. Elementary formula assignment of features was realized with in-house software NetCalc 2015 (v.1.1a, Helmholtz Zentrum Muenchen) [24]. A visual compounds–targets network was built by Cytoscape Version 3.9.0. Annotation of the correlated features were realized using online databases KEGG (KEGG: Kyoto Encyclopedia of Genes and Genomes, n.d.) and Metlin (Metlin, n.d.), and the online tool Oligonet [28].

## 4. Conclusions

In the present study, (-)ESI FT-ICR-MS-based metabolomics were combined with sensory analysis to characterize the chemical diversity of compounds related to white wine’s oxidative stability during bottle aging. Molecular markers of wine’s oxidative stability were first described by a specific CHONS chemical space in accordance with already published results indicating the increase in this specific molecular fingerprint during aging. Our findings based on the comparison of the ratios of H/C and O/C for CHOS- and CHONS-containing compounds evidenced that, in their majority, S-containing molecular features possessing O4∼12 number appeared to be related to the good state of aged wines, indicating the importance of sulfonation reactions instead of dimerization reactions on wine’s oxidative stability during aging. Moreover, molecular networking provided a more complete overview of the sulfonation molecular targets during wine aging. Indeed, based on sulfonation reaction transformations, we could report the importance of hitherto unknown and/or minor sulfur dioxide binders on wine’s oxidative stability.

## Figures and Tables

**Figure 1 metabolites-12-00323-f001:**
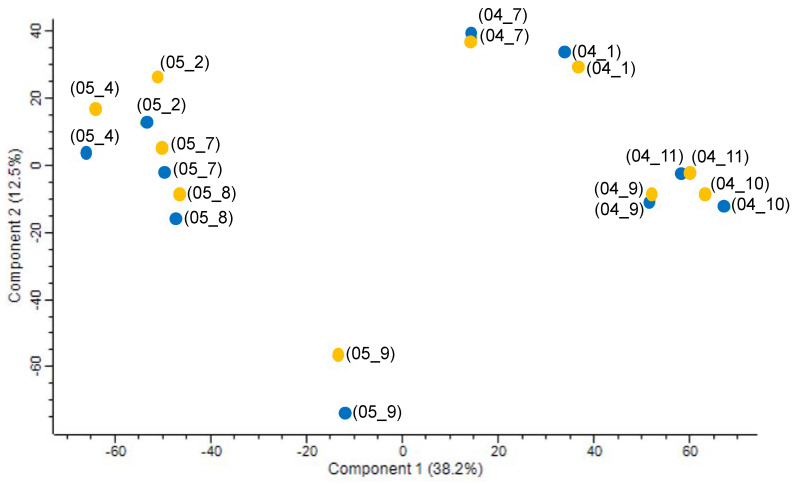
PCA of FT-ICR-MS data for the 20 bottles of wine from 2004 (04) and 2005 (05) vintages from different vineyard plots (1; 2; 4; 7; 8; 9; 10; 11), qualified sensory as oxidized (yellow) or good (blue).

**Figure 2 metabolites-12-00323-f002:**
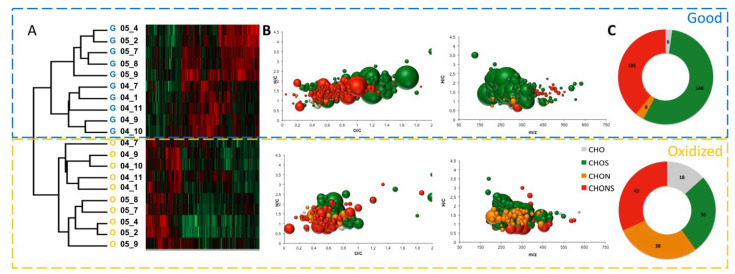
Characteristic (-)ESI FT-ICR-MS-derived molecular formulas elaborated by hierarchical cluster analysis. (**A**) Hierarchical cluster analysis of the assigned (-)ESI FT-ICR-MS-derived molecular formulas observed in wines from 2004 (04) and 2005 (05) vintages and different vineyard plots (1; 2; 4; 7; 8; 9; 10; 11), according to their oxidative character (O: oxidized; G: good). (**B**) Van Krevelen diagrams depict the most representative characteristic molecular formulas for the two main subclusters. Van Krevelen plots were colored according to molecular classes, i.e., CHO (grey), CHON (orange), CHOS (green), CHONS (red). The bubble area depicts the relative mass peak intensity within the respective subcluster. (**C**) Ring charts displaying the allocation of molecular classes.

**Figure 3 metabolites-12-00323-f003:**
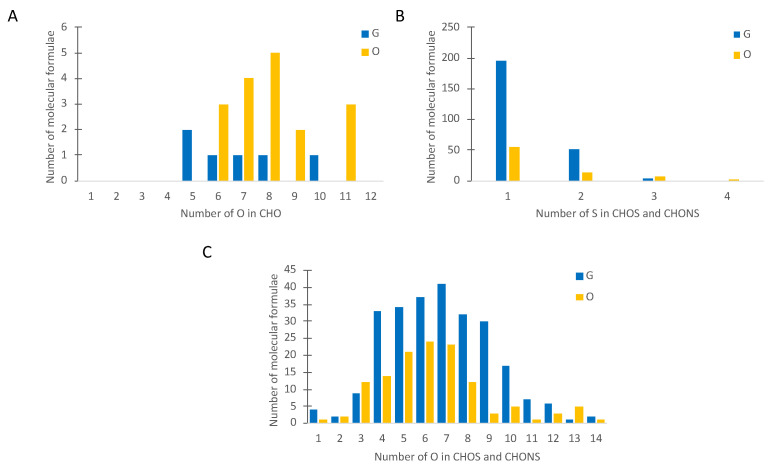
Counts of elemental formulas for both good (G) and oxidized (O) wines, with increasing numbers of O atoms in CHO (**A**), with increasing numbers of S atoms in CHOS and CHONS (**B**), and with increasing numbers of O atoms in CHOS and CHONS (**C**).

**Figure 4 metabolites-12-00323-f004:**
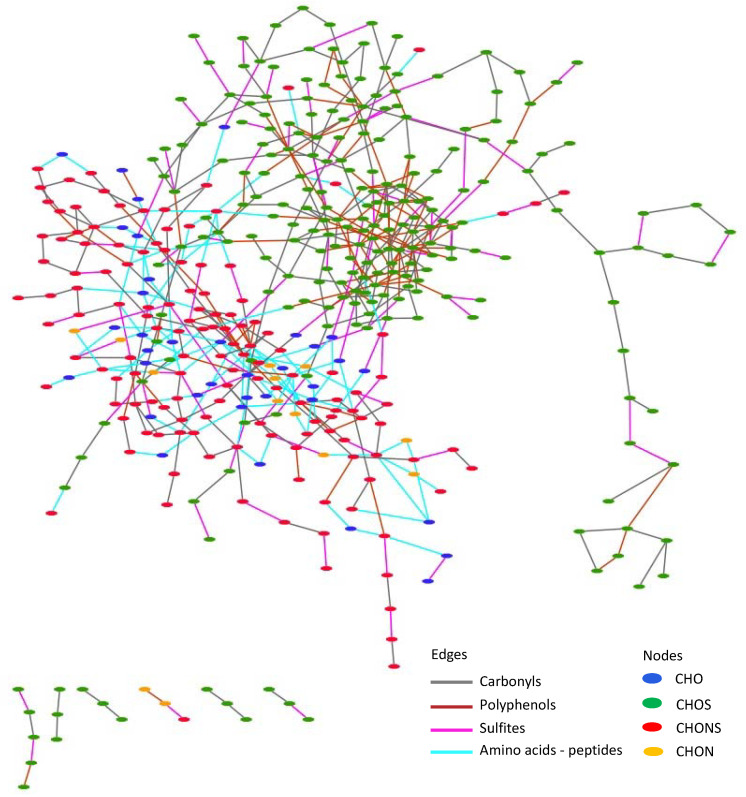
Full compositional network built from the CHONS chemical space of G wines, with nodes colored according to molecular classes, i.e., CHO (blue), CHOS (green), CHONS (red), and CHON (orange), and edges colored according to mass differences specific to nucleophilic addition reactions of known phenolic, sulfites (HSO_3_^−^), carbonyls, and amino acids—peptide compounds with unknowns in G-related molecular markers.

**Figure 5 metabolites-12-00323-f005:**
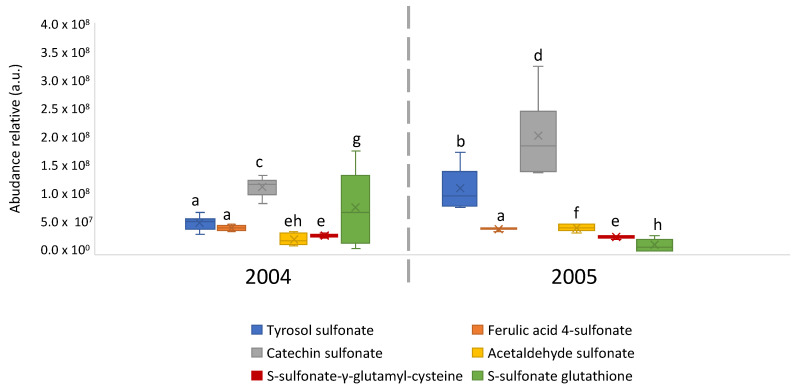
Box plots of the relative abundances of putatively identified sulfonated analogues of tyrosol, ferulic acid, catechin, acetaldehyde, γ-glutamyl-cysteine, and glutathione related to the good oxidative state (G) of aged wines from 2004 and 2005 vintages. Means not sharing a letter are significant (*p* < 0.05).

## Data Availability

The data presented in this study are available in article.

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
