# Peer review of "Sulfonation Reactions behind the Fate of White Wine’s Shelf-Life"

_metabolites, 2022, doi:10.3390/metabo12040323_

Round 1

Reviewer 1 Report

In this study, the authors evaluate the importance of sulfonation reactions in white wines' oxidative stability. This work adds new information to wine science and producers. The manuscript and the scientific work are well structured and explained. My comments are in the PDF file, marked in yellow along with the text.

Author Response

Please find attached the pdf with the responses to each highlight point.

Thank for the interest you have showed in our work.

Reviewer 2 Report

The article's topic corresponds to the thematic framework of the journal Metabolites.

It is well structured, and English is adequate.

The article provides insights into the chemical diversity of compounds related to white wine's oxidative stability during bottle aging. It represents a significant contribution to clarifying these issues in an understandable, clear, and objective way.

The quality of Figure 5 should be improved.

Author Response

Please find below the responses in green for each point.

thank for the interest you have showed in our work

It is well structured, and English is adequate.

The article provides insights into the chemical diversity of compounds related to white wine's oxidative stability during bottle aging. It represents a significant contribution to clarifying these issues in an understandable, clear, and objective way.

The quality of Figure 5 should be improved.

Figure 5 was changed. Resolution was improved and statistical analysis was added.

Reviewer 3 Report

Reviewer comment for manuscript: Sulfonation reactions behind the fate of white wines shelf-life.

I was glad to review your research; however I would like to suggest the following points to strengthen your article:

Line 13 - please review "data mimimg"

Experimental design section - given that some compounds involved in the oxidation processes of wines originate in the yeasts used for fermentation, please clarify whether the same type of yeast was used in the two years of vinification and for each sample.

Please review the name of the wine samples throughout the paper. There is no context in which it should be written in lower case.

Line 68-69. Given the findings of the research, please review the statement. The purpose of a research cannot be just to initiate questions!

Please review the contents of the bibliography carefully. The formatting must be done in full according to the requirements of the journal.

Author Response

Please find below the responses in green for each point

thank for the interest you have showed in our work

Reviewer comment for manuscript: Sulfonation reactions behind the fate of white wines shelf-life.

I was glad to review your research; however I would like to suggest the following points to strengthen your article:

Line 13 - please review "data mimimg" changed

Experimental design section - given that some compounds involved in the oxidation processes of wines originate in the yeasts used for fermentation, please clarify whether the same type of yeast was used in the two years of vinification and for each sample.

The winery uses wild yeasts for all fermentations. I am sorry but I cant provide more specifications

Please review the name of the wine samples throughout the paper. There is no context in which it should be written in lower case. Changed “O” used for oxidized samples and “G” for good (non oxidized samples)

Line 68-69. Given the findings of the research, please review the statement. The purpose of a research cannot be just to initiate questions!

Changed to: The main scope of the present study, was to examine the importance of key chemical reactions like sulfonation in wines oxidative stability during bottle ageing.

Please review the contents of the bibliography carefully. The formatting must be done in full according to the requirements of the journal.

done